Mosquitoes on a chip—environmental DNA-based detection of invasive mosquito species using high-throughput real-time PCR

Wittwer Claudia claudia.wittwer@senckenberg.de 1 2 3
Sharif Chinhda 1
Schöck Isabelle 1 2 3
Klimpel Sven 1 3 4 5
1 Institute for Ecology, Evolution and Diversity, Faculty of Biological Sciences, Goethe University Frankfurt , Frankfurt am Main , Hessen , Germany
2 Conservation Genetics Group, Senckenberg Nature Research Society , Gelnhausen , Hessen , Germany
3 LOEWE Centre for Translational Biodiversity Genomics (LOEWE-TBG) , Frankfurt am Main , Hessen , Germany
4 Senckenberg Biodiversity and Climate Research Centre (SBiK-F) , Frankfurt am Main , Hessen , Germany
5 Branch Bioresources, Fraunhofer Institute for Molecular Biology and Applied Ecology , Gießen , Hessen , Germany
Silva Daniel
Electronic publication date: 2024 Sep 30
Publication date: 2024
Volume: 12
Electronic Location ID: e17782
Received 2024 Jan 19; Accepted 2024 Jun 30
Copyright: ©2024 Wittwer et al.
Copyright year: 2024
Copyright holder: Wittwer et al.
License: This is an open access article distributed under the terms of the Creative Commons Attribution License, which permits unrestricted use, distribution, reproduction and adaptation in any medium and for any purpose provided that it is properly attributed. For attribution, the original author(s), title, publication source (PeerJ) and either DOI or URL of the article must be cited.
License URL: https://creativecommons.org/licenses/by/4.0/

Keywords: Real time PCR, High-throughput- real time PCR, Environmental DNA, Presence/absence, Biomonitoring, Invasive species

Funding: LOEWE-TBG LOEWE/1/10/519/03/03.001(0014)/52 This work was funded by LOEWE-TBG (funding code LOEWE/1/10/519/03/03.001(0014)/52). The funders had no role in study design, data collection and analysis, decision to publish, or preparation of the manuscript.

==============================
The monitoring of mosquitoes is of great importance due to their vector competence for a variety of pathogens, which have the potential to imperil human and animal health. Until now mosquito occurrence data is mainly obtained with conventional monitoring methods including active and passive approaches, which can be time- and cost-consuming. New monitoring methods based on environmental DNA (eDNA) could serve as a fast and robust complementary detection system for mosquitoes. In this pilot study already existing marker systems targeting the three invasive mosquito species Aedes (Ae.) albopictus, Ae. japonicus and Ae. koreicus were used to detect these species from water samples via microfluidic array technology. We compared the performance of the high-throughput real-time PCR (HT-qPCR) system Biomark HD with real-time PCR (qPCR) and also tested the effect of different filter media (Sterivex® 0.45 µm, Nylon 0.22 µm, PES 1.2 µm) on eDNA detectability. By using a universal qPCR protocol and only 6-FAM-MGB probes we successfully transferred these marker systems on the HT-qPCR platform. All tested marker systems detected the target species at most sites, where their presence was previously confirmed. Filter media properties, the final filtration volume and observed qPCR inhibition did not affect measured Ct values via qPCR or HT-qPCR. The Ct values obtained from HT-qPCR were significantly lower as Ct values measured by qPCR due to the previous preamplification step, still these values were highly correlated. Observed incongruities in eDNA detection probability, as manifested by non-reproducible results and false positive detections, could be the result of methodological aspects, such as sensitivity and specificity issues of the used assays, or ecological factors such as varying eDNA release patterns. In this study, we show the suitability of eDNA-based detection of mosquito species from water samples using a microfluidic HT-qPCR platform. HT-qPCR platforms such as Biomark HD allow for massive upscaling of tested species-specific assays and sampling sites with low time- and cost-effort, thus this methodology could serve as basis for large-scale mosquito monitoring attempts. The main goal in the future is to develop a robust (semi)-quantitative microfluidic-based eDNA mosquito chip targeting all haematophagous culicid species occurring in Western Europe. This chip would enable large-scale eDNA-based screenings to assess mosquito diversity, to monitor species with confirmed or suspected vector competence, to assess the invasion progress of invasive mosquito species and could be used in pathogen surveillance, when disease agents are incorporated.

Introduction

The ongoing territorial expansion of alien mosquito species poses a substantial threat to human and animal health due to a large number of potentially transmissible pathogens (Juliano & Lounibos, 2005; Medlock et al., 2012; Medlock et al., 2015; Schaffner, Medlock & Van Bortel, 2013). The invasion of non-endemic culicid species to new areas is mainly driven by globalization and climate change (Cunze et al., 2016). Early detection of non-indigenous species before invasion and establishment in new habitats allows fast and appropriate management measures (Metha et al., 2007). Thus, effective monitoring of emerging non-indigenous mosquito species is of particular importance.

In general, mosquito monitoring can be performed via active and passive approaches. Active monitoring methods involve trained personnel actively searching for and directly collecting mosquitoes in their natural habitats (Pernat et al., 2021a). These approaches either target specific life stages (eggs, larvae, pupae, adults; e.g., Djiappi-Tchamen et al., 2022; Li et al., 2020; Schneider et al., 2016; Snow & Medlock, 2008) or specific behavioral traits (e.g., blood-feeding, Brugman et al., 2017). Alternatively, traps can be deployed to attract and/or capture mosquitoes. A variety of trap types exist either targeting adult mosquitoes with baited traps (e.g., Becker et al., 2010; Gorsich et al., 2019) or providing preferential resting or egg deposition sites with oviposition traps (e.g., Brugman et al., 2017; Djiappi-Tchamen et al., 2022; Johnson, Ritchie & Fonseca, 2017). Passive monitoring approaches, on the other hand, rely on citizens’ collaboration (citizen science) to obtain information about the current species composition and abundance, the distribution range (Eritja et al., 2021; Jordan, Sorensen & Ladeau, 2017; Kampen et al., 2015; Pernat et al., 2021a) and the transmission risk of pathogens (Murindahabi et al., 2021).

The application of both active and passive monitoring methods led to the first evidence of the occurrence of three alien mosquito species Aedes (Ae.) albopictus, Ae. japonicus and Ae. koreicus in Germany (Friedrich-Loeffler-Institut , 2019). These mosquito species are considered to be the most important alien mosquito species in Europe (Martinet et al., 2019) due to their confirmed and assumed vector competence for a variety of viruses (e.g., Ciocchetta et al., 2018; Grard et al., 2014; Jansen et al., 2018; La Ruche et al., 2010; Little et al., 2021; Rezza et al., 2007; Schaffner et al., 2011; Schneider et al., 2016). Whereas Ae. koreicus and Ae. japonicus usually breed in artificial or natural containers such as grave vases or tree holes in rural areas, Ae. albopictus can breed in an even wider variety of water reservoirs, making it highly adaptable to sub-/urban environments (Bartlett-Healy et al., 2012; Becker et al., 2010; Li et al., 2014; Montarsi et al., 2015; Sáringer-Kenyeres, Bauer & Kenyeres, 2020). The continuous expansion of these mosquito species to new areas raises concerns for public health authorities, prompting the need for action to implement efficient vector surveillance and control measures to reduce the transmission risk of the associated pathogens (ECDC, 2012; World Health Organization, 2021).

Although active and passive monitoring techniques can provide comprehensive monitoring data to address these issues, the complete monitoring process, from sampling to species identification, can be complex. Active approaches are often laborious and costly due to the requirement to recruit skilled personnel for accurate sampling (Moise et al., 2019) and morphological identification (Jourdain et al., 2018) or the need to purchase and maintain suitable trapping devices (Palmer et al., 2017). Additionally, the outcomes of trap-based monitoring methods can be inconsistent due to the influence of biotic (e.g., oviposition behavior and timing, Früh et al., 2020; Silva et al., 2021) and abiotic environmental factors (e.g., weather conditions, Gorsich et al., 2019) as well as factors related to the trapping process such as trap design, placement, attractant efficacy, and maintenance (Gorsich et al., 2019; Reisen et al., 1999; Sikaala et al., 2013). Compared to active approaches, passive monitoring based on citizen science is more cost-effective (Braz Sousa et al., 2020; Palmer et al., 2017). Yet, relying solely on the data collections generated by volunteers can have adverse effects on accurate and reliable species detection, mainly due to the variability of data collection methods (Braz Sousa et al., 2022) and data quality (Bird et al., 2014) as well as data collection biases arising from spatial, temporal and demographic factors (Dekramanjian et al., 2023; Palmer et al., 2017; Pernat et al., 2021b; Weiser et al., 2020).

Due to the challenges inherent in the application of conventional monitoring, researchers are increasingly turning to innovative monitoring approaches based on environmental DNA (eDNA), which could be a promising tool to complement current mosquito surveillance techniques. eDNA methodology for species monitoring allows for the detection of target species without the need for sampling tissues, solely relying on environmental samples such as water, soil or faeces (Taberlet et al., 2012a; Thomsen & Willerslev, 2015). In the aquatic environment, eDNA-based detection was shown to be suitable for diverse species such as fishes (e.g., Jerde et al., 2011; Spens et al., 2017; Takahara et al., 2012), amphibia (e.g., Bálint et al., 2018; Ficetola et al., 2008; Goldberg et al., 2011), invertebrates (e.g., Agersnap et al., 2017; Thomsen et al., 2012; Tréguier et al., 2014) and pathogens (e.g., Wittwer et al., 2018). Since mosquitoes require aquatic environments for complete metamorphosis, the eDNA approach seems to be a promising candidate for Culicidae biomonitoring. eDNA was shown to persist in water only for up to several weeks (Barnes et al., 2014; Dejean et al., 2011), with one study reporting mosquito eDNA being detected up to 25 days (Schneider et al., 2016). Schneider et al. (2016) were the first to show the suitability of eDNA-based mosquito detection for the invasive species Ae. albopictus, Ae. japonicus and Ae. koreicus in water samples. Their qPCR approach revealed a higher detection probability compared to eDNA metabarcoding and traditional survey methods such as larval sampling. Although this method already accurately detects the target species from water samples, these single-species qPCR approaches are complicated to scale up for large-scale surveillance. eDNA metabarcoding, on the other hand, allows the simultaneous detection of multiple organisms via next-generation sequencing approaches using short taxon-specific eDNA markers (Batovska et al., 2018; Boerlijst et al., 2019; Gutiérrez-López et al., 2023; Pedro et al., 2020; Schneider et al., 2016; Taberlet et al., 2018). eDNA metabarcoding enables comprehensive biodiversity assessments of mosquitoes (Gutiérrez-López et al., 2023) within an ecosystem and can be cost-effective, when analyzing multiple samples simultaneously (Bálint et al., 2018; Batovska et al., 2018; Lafferty, 2024).

Another detection method with high potential to further increase the number of analyzed eDNA samples, together with a high number of species-specific eDNA markers for diverse mosquito species in parallel, could be microfluidic platforms for High-Throughput qPCR, such as the Biomark HD (Standard BioTools Inc., South San Francisco, CA, USA). The Biomark HD system utilizes Integrated Fluidic Circuits (IFCs) as microfluidic devices, which enable to perform up to 9,216 individual qPCR reactions simultaneously. Recently, the Biomark HD qPCR system proofed suitable for the use in environmental DNA analysis and biodiversity monitoring (e.g., Shea et al., 2022; Van der Pouw Kraan et al., 2024). eDNA chips based on Biomark HD IFCs targeting multiple culicid species at once could be a reasonable complement to conventional and other eDNA-based monitoring methods due to the ease, the rapidity of data acquisition, and the possibility to perform quantitative analysis similarly to conventional qPCR methodologies. To reach this, specific and sensitive culicid marker systems need to be developed and optimized for the use on the Biomark HD qPCR system, along with the test and integration of already established qPCR assays. To date, qPCR assays applicable for eDNA analysis exist for Ae. albopictus (Hill et al., 2008), Ae. japonicus (Hill et al., 2008; Van de Vossenberg et al., 2015), Ae. koreicus (Schneider et al., 2016) and diverse other mosquito species (Kristan et al., 2023; Mee et al., 2021; Odero et al., 2018; Sakata et al., 2022). Since the used qPCR chemistries and protocols considerably vary, the qPCR conditions of all assays need to be standardized for the use on Biomark HD IFCs for a parallel reliable and accurate detection. Besides the standardization of eDNA markers, also the eDNA sampling and extraction schemes need to be harmonized to increase the detection probability of all possibly occurring species in a given habitat (Barnes & Turner, 2016; Goldberg et al., 2016; Lodge et al., 2012; Rees et al., 2015; Taberlet et al., 2012b). The water sampling procedures for eDNA-based mosquito detection include precipitation approaches of small water volumes up to 40 ml (Gutiérrez-López et al., 2023; Kristan et al., 2023; Odero et al., 2018; Schneider et al., 2016) and filtration methods, which allow for the processing of larger water volumes being concentrated on filters (Boerlijst et al., 2019; Sakata et al., 2022), thus enhancing the detection probability of rare and recently introduced species occurring at low population densities (Mächler et al., 2016).

The present study was set out as a pilot study to detect the three mosquitoes Ae. albopictus, Ae. japonicus and Ae. koreicus from eDNA samples by transferring the already existing eDNA markers for these species from Hill et al. (2008), Van de Vossenberg et al. (2015) and Schneider et al. (2016) to the high-throughput qPCR platform Biomark HD as a first step towards an eDNA chip targeting all mosquito species occurring in Western Europe. Along with the standardization of the qPCR chemistry and protocol to allow for a reliable and parallel detection of all target organisms, we also evaluated the effect of different filter media and pore sizes (Sterivex® 0.45 µm, Nylon 0.22 µm, PES 1.2 µm) on eDNA detectability by assessing the performance of the qPCR versus HT-qPCR system in terms of presence/absence measurements and inhibition.

Materials & Methods

Site selection for reference specimen and eDNA analysis

In total 14 sites with known occurrences of the target species were selected in collaboration with KABS e.V. (Table S1). For the eDNA-based detection of Ae. japonicus and Ae. koreicus sampling sites in and around the city of Wiesbaden were selected. The Friedhofsverwaltung Wiesbaden and Friedhofsverwaltung Ginsheim gave permission to sample on cemeteries. Although no comprehensive parallel conventional mosquito monitoring was performed alongside the eDNA sampling, established populations of Ae. japonicus and Ae. koreicus were assumed in these areas due to continuous monthly measurements conducted by KABS e.V. members since many years (D. Reichle, KABS e.V., 2022, pers. comm.). eDNA sites for Ae. albopictus detection were chosen and sampled by collaborators from KABS e.V., before eradication measures were deployed.

Larval reference specimens of Ae. japonicus and Ae. koreicus were collected on cemeteries from May to September 2020. Reference specimens of Ae. albopictus were obtained from laboratory reared larvae from Biogents AG (Regensburg, http://www.biogents.com). Mosquito larvae from the field were morphologically identified in the laboratory using a binocular microscope and mosquito identification keys (Becker et al., 2010; Tanaka, Mizusawa & Saugstad, 1979).

Analysis steps for reference specimen

DNA extraction

DNA of morphologically identified larvae was extracted for positive controls and standard curve construction. Initially, larvae were decapitated to diminish potential inhibition during PCR amplification (Beckmann & Fallon, 2012). The extractions were performed using the DNeasy Blood & Tissue Kit (Qiagen GmbH, Hilden, Germany) following the manufacturer’s protocol with modifications including lysis for 24 h at 56 °C and 350 rpm. Genomic DNA was finally eluted in 80 µl AE buffer and stored at −20 °C.

Molecular species identification (DNA barcoding)

Mosquito larvae from the field were also genetically identified based on the cytochrome c oxidase subunit I (COI) gene. A ∼425 bp fragment was amplified using primers C1-J-1718 (forward: 5′-GGAGGATTTGGAAATTGATTAGTTCC-3), C1-N-2191 (reverse: 5′-CCCGGTAAAATTAAAATATAAACTTC-3′; Simon et al., 1994) and the Mastercycler® nexus (Eppendorf AG, Hamburg, Germany). Each reaction of 25 µl contained 2 µl DNA extract, Qiagen Taq PCR Master Mix (2.5 units Taq DNA Polymerase, 1.5 mM MgCl2, 200 µM of each dNTP), 25 nM of each primer. The thermal profile was the following: 5 min at 95 °C, 35 cycles of 30 s at 94 °C, 40 s at 51 °C and 45 s at 72 °C, followed by a final extension step at 72 °C for 10 min. Amplification products were separated by agarose gel electrophoresis and visualized on a 1.5% agarose gel stained with GelRed® Nucleid Acid Gel Stains (Biotium, Fremont, CA, USA). DNA purification and Sanger sequencing were performed by Microsynth Seqlab (Microsynth Seqlab GmbH, Göttingen, Germany). Every sequence was checked with the Ugene tool (Okonechnikov, Golosova & Fursov, 2012) and was compared with sequences deposited in the NCBI GenBank using the BLAST algorithm (Altschul et al., 1997).

Analysis steps for eDNA samples

eDNA sampling

eDNA samples (Fig. 1) of Ae. japonicus and Ae. koreicus sites were collected on 30.08.2022 from surface water (Boerlijst et al., 2019) at multiple random open water basins at the cemeteries Wiesbaden (WI) Südfriedhof, WI Naurod, WI Igstadt and Ginsheim. Initially water samples were prefiltered with 100 µm pore sized nylon filters into 5L-canisters to exclude larger vegetational components. Subsequently the prefiltered water was processed on three different filter media with varying pore sizes (Sterivex® 0.45 µm, Nylon 0.22 µm, PES 1.2 µm), each adjusted to a total of 200 ml filtrated water measured with a beaker (Table S1). Two filters per site and per filtration method were produced with a portable peristaltic pump (Masterflex, Cole-Palmer Instrument Company, LLC, Chicago, IL, USA). Additionally, field blank controls of each filter type were produced. Filters were either transferred to a 5 ml Eppendorf tube (Nylon, PES) or sealed with caps in plastic bags (Sterivex®) and placed in an icebox during the field sampling. Later the same day the filters were stored in laboratory freezers at −20 °C until further processing. Decontamination procedures included thorough cleaning of inner and outer parts of the filtration apparatus and equipment with 20% bleach solution as well as deionized water (DI).

Aedes albopictus sites were sampled by collaborators of KABS e.V. solely with Sterivex® filters, until the filters were clogged. Sterivex® filters were also used for Ae. albopictus breeding water samples at maximum filtration capacity and served as positive eDNA controls (Table S1).

eDNA extraction

The eDNA extraction was performed by experienced lab personal in a room dedicated to non-invasive sample processing. Sterivex® eDNA samples were extracted according to Sigsgaard et al. (2016) modified by Riaz et al. (2023). eDNA from PES and Nylon filters were extracted according to the following protocol: frozen filters were allowed to thaw at RT for approximately 45 min. After discarding the white edges, the filters were quartered and 2 14 parts per sample were used for further analysis. Each quarter was transferred to a Lyse&Spin Basket column (Qiagen GmbH, Hilden, Germany) placed in a 2 ml reaction tube. In each column 360 µl ATL (Blood & Tissue Kit, Qiagen GmbH, Hilden, Germany) and 40 µl Proteinase K was added. The samples were incubated at 56 °C for 3 h while permanently shaking. The lysates of the quarter pieces were pooled, mixed again and centrifuged. A total of 600 µl of combined lysate was used for further processing in an automated QIAcube DNA extraction system (Qiagen GmbH, Hilden, Germany) with a slightly modified extraction protocol “Human Stool DNA Analysis”, including two elution steps of 30 µl AE elution buffer. The eDNA samples were stored at −20 °C until further processing. One extraction blank control was included during the extraction procedure.

Figure 1 Study scheme of this study.

eDNA samples of Ae. japonicus and Ae. koreicus sites were collected at multiple random open water basins at four cemeteries in Hessen with three different filter media with varying pore sizes, whereas Ae. albopictus sites were sampled at divers water reservoirs and breeding water solely with Sterivex® filters until clogging. After eDNA extraction the samples where analyzed via Real-Time PCR and high-throughput real-time PCR on a Biomark HD platform with 96.96 Dynamic IFCs. Created with BioRender.com with mosquito illustrations from J. L. Ordóñez.

Real-Time PCR

For the detection of the three mosquito species developed and tested species specific and sensitive qPCR assays in previous studies (Hill et al., 2008; Schneider et al., 2016; Van de Vossenberg et al., 2015) were used with 6-FAM-labelled TaqMan MGB-NFQ probes to adjust the qPCR assays for the Biomark HD system. qPCR was performed on a Quantstudio3 Real-Time-PCR system (Applied Biosystems, Waltham, MA, USA) and Ct values were calculated with the QuantstudioTM Design and Analysis software (version 1.5.1; Applied Biosystems, Waltham, MA, USA). In a final reaction volume of 10 µl qPCR reactions were composed of 5 µl 2xTaqMan Environmental Master Mix 2.0 (Thermo Fisher Scientific, Waltham, MA, USA), 0.5 µM of each primer, 0.2 µM of probe, 2 µl DNA template and 1.8 µl of molecular-grade water. The qPCR program consisted of an initial Taq polymerase heat activation at 95 °C for 10 min followed by 45 cycles at 95 °C for 15 s and 62 °C for 30 s. All eDNA samples were run in triplicate. Also biological reference specimen samples at 10 ng/µl serving as positive control as well as NTCs (PCR water), field blank and extraction blank controls of each filter type were included in duplicate on each plate. For standard curve construction and sensitivity testing the genomic DNA of reference specimen was used. For all qPCR assays a 10x dilution series was created from 10 ng/µl to 0.01 pg/µl based on initial Nanodrop® measurements. Each dilution was assessed 10 times to determine assay-specific qPCR efficiency, R2 and LOD/LOQ. To screen for filter-dependent PCR inhibition one eDNA sample per filter type for Ae. japonicus and Ae. koreicus at site cemetery WI Igstadt was analyzed via qPCR with TaqMan Exogenous Internal Positive Control (IPC) Assay (Life Technologies, Carlsbad, CA, USA). PCR inhibition is indicated by a Ct value shift of ≥ 1 in the IPC assay relative to control samples with 1xTE (Wilcox et al., 2020).

Preamplification and high-throughput real-time PCR

The targeted preamplification was performed in 96-well plates on a T1 Thermocycler (Biometra, Gottingen, Germany). The 10 µl PCR reactions consisted of 5 µl 2xTaqMan Environmental Master Mix 2.0 (Thermo Fisher Scientific, Waltham, MA, USA), 1 µl of 10x STA primer mix (equimolar concentration of all forward and reverse primers at 500 nM each), 2 µl DNA template and 2 µl of molecular-grade water. The PCR protocol included heat activation at 95 °C for 10 min and 28 cycles of 15 s at 95 °C denaturation and 3 min at 62 °C annealing/elongation followed by a cooling step at 6 °C. After preamplification the samples were diluted 1:10 and analyzed on a Biomark HD system (Standard BioTools Inc., South San Francisco, CA, USA) with 96.96 Dynamic IFCs. Priming and sample/assay loading was performed on an IFC Controller HX with the 136x script. The loaded IFCs were run in the Biomark HD with following protocol: thermal mix at 70 °C for 40 min and at 60 °C for 30 s followed by hot start at 95 °C for 10 min and 35 cycles at 95 °C for 15 s and at 62 °C for 30 s. Each biological sample was analyzed in duplicate on two independent IFCs with 24 and 31 technical replicates respectively. Both IFCs included tissue samples of the target species at 0.01 ng/µl as positive controls. An additional IFC included the following negative controls in duplicate: a NTC (PCR water), field blank controls and extraction blank controls of each filter type. HT-qPCR data collection was performed using the Biomark Data Collection software (Fluidigm Corp., South San Francisco, CA, USA). Data analysis and vizualisation was conducted with the Fluidigm Real-Time PCR analysis software (version 4.3.1., Fluidigm Corp.). The analysis settings were set to a quality threshold of 0.50, the baseline correction method to “Linear (Derivative)” and the Ct threshold method to “Auto (Global)”. A result was counted as successful, when a target species was detectable in at least one biological replicate on both IFCs with a quality value >0.50.

Target sequence confirmation via M13-tagged primers and Sanger sequencing

Forward and reverse primers of all tested mosquito species were tagged with M13 sequences (M13F-tag: TGTAAAACGACGGCCAGT; M13R-tag: CAGGAAACAGCTATGAC) at the 5′-end to verify the eDNA results obtained with qPCR and HT-qPCR. Per sampling site one sample was analyzed with the given qPCR protocol with M13-tagged primers and additional probe to ensure successful amplification. The qPCR reactions were cleaned with Exosap, diluted with nuclease-free water and sequenced via Sanger sequencing on an ABI 3730 DNA analyzer (Applied Biosystems, Waltham, MA, USA). The sequencing results were analyzed with SeqScanner2.0 (Applied Biosystems, Waltham, MA, USA) and compared with NCBI GenBank sequences using BLAST to check for correct species assignment. Raw reads of eDNA mosquito sequences are obtainable from the NCBI Sequence Read Archive (SRA) database (BioProject accession number PRJNA1064692).

Statistics

GraphPad Prism, version 10.2.2 (GraphPad Software Inc., La Jolla, CA, USA, 2024) was used for further data analysis and graphical visualization. Normal distribution of measured Ct values was evaluated with a Shapiro–Wilk normality test and results were plotted as QQplot (Fig. S1). Due to non-normal distribution of data nonparametric Mann–Whitney-U tests were performed on measured Ct values of all positive sampling sites to evaluate the filter- and platform-specific performance. For comparing the eDNA detectability as a function of water filtration volume, a linear regression was performed on measured Ct values of Ae. albopictus sites. A linear regression as well as Spearman R correlation analysis was performed on the obtained Ct values of both qPCR methods across all positive sampling sites and all tested organisms to test for the interdependence of measured Ct values. Outlier identification was performed with the ROUT method (Q = 1%). Post-hoc tests on statistical power were carried out with G*Power (Version 3.1.9.7). Statistical significance was set at α = 0.05.

Results

Assay specificity and sensitivity

In this study, qPCR amplification of all mosquito species was successful with 6-FAM-MGB probes for all assays and one universal qPCR protocol. In vitro testing with reference tissue samples of target/non-target species (Table S2) showed that the markers amplified the target species with a measured Ct value of 16.54 for Ae. albopictus, 33 for Ae. japonicus and 23.84 for Ae. koreicus at a DNA concentration of 10 ng/µl. The Ae. albopictus assay also amplified Ae. koreicus-DNA at a very high Ct-value of 37. Since eDNA samples have much lower DNA content this co-amplification will not hamper specific species detection in real-world samples. All other tested reference specimens of culicid species failed to amplify along with all negative controls and human DNA controls. The qPCR efficiency was determined at 90% for the Ae. albopictus assay, 99% for the Ae. japonicus assay and 94% for the Ae. koreicus assay with calculated LOD and LOQ at concentrations of 1 pg/µl and 0.1 pg/µl tissue DNA, respectively (Figs. S2, Table S3). Selected eDNA samples and reference tissue samples were Sanger-sequenced with M13-tagged primers and confirmed the presence of target sequences DNA at sites previously tested positive via qPCR/HT-qPCR (Table S4).

eDNA detectability with different filter media and qPCR method

Aedes japonicus was detected at the cemetery WI Igstadt with all filter types with average Ct values for qPCR at 38.34 ± 1.31 and HT-qPCR at 11.69 ± 0.64 (Fig. 2, Table 1). Aedes koreicus-eDNA was detected with all tested filters at cemeteries WI Igstadt (qPCR Ct 38.49 ± 1.23; HT-qPCR Ct 11.33 ±0.31) and WI Südfriedhof (qPCR Ct 37.13 ± 0.43; HT-qPCR Ct 9.46 ± 0.41). For the positive Ae. japonicus and Ae. koreicus sampling sites the observed Ct values did not vary significantly between filter media. PCR inhibition tests for all filter media at sampling site “cemetery WI Igstadt” revealed marked inhibition in PES samples with Ct shifts of 17.4 and Sterivex® of 4.7 for the Ae. japonicus assay. Also the Ae. koreicus assay showed increased Ct shifts of 3.6 for PES filters and 2.1 for Sterivex® filters. Nylon filters did not show an inhibited amplification for both assays (Ae. japonicus: 0.56; Ae. koreicus: 0.15; Fig. S3). The Ae. koreicus and Ae. japonicus assays failed to amplify in all Ae. albopictus eDNA breeding water samples with both qPCR methods. Inconclusive results were obtained for both assays on eDNA sampling site “cemetery WI Naurod”, when amplification was solely observed on the Biomark HD platform with different filters and/or technical replicates.

Figure 2 Results of eDNA samples of Ae. japonicus and Ae. koreicus achieved with different filter types (PES, Sterivex® and Nylon).

Obtained Ct values were measured via qPCR (●) and HT-qPCR (■) with site- and method- specific average Ct ± SD values.

Table 1 Comparison of obtained Ct values measured with qPCR and HT-qPCR.

Each eDNA sample was tested with the eDNA assays targeting Ae. albopictus, Ae. japonicus and Ae. koreicus with qPCR and HT-qPCR. Mean Ct values per site, species and filter medium are given. The signal strength is indicated by measured mean Ct values with lower Ct-value implying a stronger eDNA signal. Slash denotes the absence of DNA in collected water samples. Asterisks (*) denote nonreproducible chip results.

No.	Sampling location		Species	qPCR
ØCt value	HT-qPCR
ØCt value	
				PES
1.2 μ m	Sterivex®
0.45 μ m	Nylon
0.22 μ m	PES
1.2 μ m	Sterivex®
0.45 μ m	Nylon
0.22 μ m	
1	Cemetery WI Südfriedhof	HE	Ae. japonicus	/	/	/	/	/	/	
			Ae. koreicus	36.8 ± 2.87	36.98 ± 1.04	37.61 ± 0.07	9.46 ± 0.4	9.05 ± 0.24	9.87 ± 0.03	
2	Cemetery WI Igstadt	HE	Ae. japonicus	39.86 ± 0.63	37.55 ± 1.05	37.62 ± 1.72	11.59 ± 0.56	11.1 ± 0.43	12.38 ± 2.44	
			Ae. koreicus	38.71 ± 0.25	39.62 ± 1.42	37.14 ± 0.03	11.45 ± 0.62	10.98 ± 0.67	11.56 ± 1.02	
3	Cemetery WI Naurod	HE	Ae. japonicus	/	/	/	/	22.24*	15.30 ± 0.49*	
			Ae. koreicus	/	/	/	14.42 ± 0.74*	13.21 ± 1.39*	/	
4	Cemetery Ginsheim	HE	Ae. japonicus	/	/	/	/	/	/	
			Ae. koreicus	/	/	/	/	/	/	
5	Open container, Ketsch	BW	Ae. albopictus		35.63 ± 4.86			3.96 ± 0.17		
6	Cable duct, Ketsch	BW	Ae. albopictus		33.94 ± 1.16			6.81 ± 1.67		
7	Watering can, Ketsch	BW	Ae. albopictus		34.71 ± 0.56			6.28 ± 0.01		
8	Rainwater tank, Bürstadt-Bobstadt	HE	Ae. albopictus		30.58 ± 0.71			6.39 ± 0.44		
9	Grave vase, Bürstadt	HE	Ae. albopictus		28.98 ± 0.21			3.89 ± 0.01		
10	Laboratory breeding water 1	BW	Ae. albopictus		37.11 ± 0.57			11.42 ± 0.84		
11	Laboratory breeding water 2	HE	Ae. albopictus		28.37 ± 0.7			3.02 ± 0.02		
12	Laboratory breeding water 3	BW	Ae. albopictus		38.56 ± 2.38			11.03 ± 0.59		
13	Laboratory breeding water 4	BW	Ae. albopictus		27.17 ± 0.14			3.22 ± 0.03		
14	Laboratory breeding water 5	BW	Ae. albopictus		28.17 ± 0.37			3.83 ± 0.71		

Aedes albopictus was detected with Sterivex® filters at all sampling sites and in all laboratory breeding water samples via qPCR with an average Ct value of 32.32 ± 4.14 and HT-qPCR with 5.99 ± 3.1 across all sites (Fig. 3). Final filtration volumes varied between 180–690 ml dependent on suspended matter (Table S1). Despite a wide variety of reached water volumes of Ae. albopictus Sterivex® samples these volumina did not affect observed Ct values via qPCR or HT-qPCR (Fig. S4). The Ae. albopictus assay did not amplify in all other eDNA samples, where its occurrence was previously excluded, with both analysis platforms.

Figure 3 Results of eDNA samples of Ae. albopictus. with obtained Ct values measured via qPCR (●) and HT-qPCR (■) stating average Ct ± SD over all sampling sites.

Over all positive sampling sites and target species the measured Ct values via HT-qPCR were significantly lower with a mean Ct of 8.28 ± 3.4 compared to mean Ct values obtained via qPCR with 35 ± 4.2 (p < 0.0001; unpaired, two-tailed Mann–Whitney-U-test). Still, the observed Ct values of both qPCR methods were significantly correlated (linear regression: p < 0.0001, Fig. 4; Spearman R value 0.8772, p < 0.0001). All NTCs, field and extraction blank results were negative and no outliers were found in the data set.

Figure 4 Comparison of achieved Ct values of both qPCR methods over all sampling sites.

Discussion

To the best of our knowledge, this is the first eDNA-based approach using a microfluidic HT-qPCR platform for mosquito detection. Due to their confirmed and assumed vector competence for a wide range of potentially life-threatening viruses the species targeted in this pilot study are the most important alien mosquito species in Europe (Martinet et al., 2019). Thus, the development and implementation of a new detection method is required, which is (1) less time-consuming, (2) more cost-effective and (3) applicable for large-scale monitoring, particularly suitable for these Aedes species.

Our results confirm that the qPCR assays for Ae. albopictus, Ae. japonicus and Ae. koreicus originally designed by Hill et al. (2008), Van de Vossenberg et al. (2015) and Schneider et al. (2016) work robustly with our modifications using 6-FAM-MGBNFQ-Probes and a universal qPCR protocol for all assays on both qPCR platforms. Although these modifications appear to result in reduced sensitivity for all qPCR assays of two to three orders of magnitude compared to Schneider et al. (2016), the overall qPCR efficiency does not seem to be impaired. All in all, the HT-qPCR analysis led to a highly robust and reproducible detection compared to single-species qPCR. Our results show that obtained Ct values via qPCR were significantly higher compared to measurements via HT-qPCR. This outcome is not surprising, since the preamplification step drastically increases the number of qPCR amplicons and thus decreases measured Ct values on HT-qPCR, but in a highly correlated manner compared to single-species qPCR. In sum, all tested filter types were able to detect the target species at sampling sites, where their occurrence was previously confirmed. Filter media properties, the final filtration volume and observed qPCR inhibition did not affect measured Ct values via qPCR or HT-qPCR. These findings are in line with the observations of previous studies using different water sampling approaches (e.g., Boerlijst et al., 2019; Kristan et al., 2023; Schneider et al., 2016). Thus, we assume that generally not much water is needed for successful eDNA-based mosquito detection, which makes the standardization of eDNA sampling schemes for Culicidae quite simple.

Despite these promising outcomes, this study involves certain limitations, which should be addressed in future studies. Due to the currently low abundance of Ae. japonicus and Ae. koreicus in the study area, the number of sites and samples is too small to infer the optimal water filtration method from the obtained results. Moreover, inconclusive results were observed for the Ae. japonicus and Ae. koreicus assays at the sampling sites WI Naurod and Ginsheim, which were not reproducible on both chips and only single filters amplified. Additionally, our results showed a disagreement between qPCR and HT-qPCR results concerning possible false positive detections. Aedes japonicus was detected at Ae. albopictus field eDNA sites, namely site 5/container, site 7/watering can and site 9/grave vase, on both eDNA chips with high Ct values (i.e., low eDNA concentration), but not via qPCR (Data S1). As proved by M13 sequencing at site 5/container these results are likely based on a previously unknown co-occurrence of both species at these sites. We assume that these non-reproducible results could originate from the presence of inhibitors possibly hampering amplification during qPCR, sensitivity and specificity issues of the used assays, qPCR amplification bias and/or primer dimer formation during the preamplification process or cross-contamination during qPCR setup and chip handling. Future assessments should focus on maximizing reproducibility by further optimizations of the used eDNA marker systems and the development of standardized eDNA laboratory protocols. Additionally, further tests should include the evaluation of other water body types such as lakes and ditches to get more insights into the effects of varying water chemistries and more complex invertebrate communities. Another aspect for future research is the validation of the specific eDNA release patterns of mosquitoes. Despite much research to assess eDNA detectability of invertebrate species (e.g., Thomsen et al., 2012; Tréguier et al., 2014), little is known of eDNA shedding rates of culicid species. Aquatic invertebrates discontinuously shed particulate DNA into water, such as exoskeletons and exuviae (Dunn et al., 2017; Tréguier et al., 2014; Watts et al., 2005). Culicid species occur in masses at certain species-specific temperature optima during the year and after heavy rainfall in temporary water bodies. During mass occurrences mosquito larvae shed vast numbers of exuviae into the breeding water during their development. High numbers of shedded juvenile exuviae can serve as important source for eDNA (Watts et al., 2005) when accumulating in stagnant water bodies, thus increasing the measurable eDNA concentration of mosquitoes in water. Moreover, laboratory experiments with Cx. pipiens pallens revealed high eDNA concentrations directly after larval hatching and pupal emergence (Sakata et al., 2022). Successful detection of mosquito eDNA in the field can be potentially hampered by a variety of ecological factors such as unfavorable weather conditions, species-specific seasonal and spatial occurrences, and suboptimal physico-chemical water parameters (e.g., too high water temperatures, UV-B-light, acidic pH; Seymour et al., 2018; Strickler, Fremier & Goldberg, 2015), which might result in increased eDNA degradation rates and thus lowering eDNA detection probability.

All in all, our study underscores the importance of standardized field and laboratory protocols as a major step towards an eDNA mosquito chip incorporating all mosquito species occurring in Western Europe. Our observations are an important contribution for the use of already existing marker systems as well as for the development of new markers targeting mosquito species to reach this goal. Recently, new qPCR assays for eDNA analysis were established for diverse culicid genera such as Culex sp. (Sakata et al., 2022), Aedes sp. (Mee et al., 2021) and Anopheles sp. (Kristan et al., 2023; Odero et al., 2018). Moreover, further assays targeting mosquito species are currently under development. Most currently available eDNA markers primarily target invasive species with a known vector competence for various pathogens. Yet, little is known about the vector competence of native European mosquitoes (Martinet et al., 2019; Vogels et al., 2017). For example, Central European Culex species are known to transmit various viruses including WNV, Sindbis and Usutu virus (Jöst et al., 2011; Nikolay, 2015), whereas Cx. pipiens (both biotypes pipiens and molestus) as well as Cx. torrentium have shown to have no vector competence for ZIKV (Heitmann et al., 2017). As infection and transmission probabilities of indigenous mosquitoes is still subject of ongoing research, we propose that marker systems should be in place in the near future to monitor these species as well. By standardization of qPCR chemistries and protocols of established and/or modified as well as newly developed markers, they will be well-suited for the use on HT-qPCR platforms for simultaneous detection of various mosquito species with water samples. Since HT-qPCR platforms such as Biomark HD allow for massive upscaling of species-specific assays and sampling sites, this methodology enables large-scale mosquito surveillance and at the same time reduces time and cost for robust species detection.

As a matter of principle one could argue to just use eDNA metabarcoding instead of facing the huge task to develop multiple species-specific eDNA marker systems for culicid species. In fact, its application may be generally easier to implement, since commonly available standard laboratory equipment is used and well-established laboratory protocols are available. Despite these advantages, this approach can introduce biases leading to incorrect species assignment, including PCR amplification bias favoring some taxa over others (Clarke et al., 2014), or insufficient taxonomic resolution of metabarcoding markers, which can hamper the differentiation between closely related species (Taberlet et al., 2012b). Moreover, common restraints in using eDNA metabarcoding methodology lie in the complex analysis of sequencing data, which requires bioinformatic expertise. Yet, with ongoing research and the use of artificial intelligence (AI) eDNA metabarcoding approaches will become more user-friendly and facilitate the analysis and interpretation of huge metabarcoding data sets (Zhang et al., 2023). The application of microfluidic platforms, on the other hand, requires expertise in marker development and specialized equipment for chip preparation and Real-Time PCR analysis, which increases the initial costs. The cost-benefit efficiency of microfluidic devices turns to advantage, once species-specific eDNA marker systems and universal qPCR protocols are developed and optimized. Since the data produced by Biomark IFCs is based on fluorescence measurements similarly to single-species qPCR, data analysis and interpretation is comparably easy, fast and highly reproducible. An additional advantage of IFCs is the need of very small amounts of fluids in the nanoliter range, resulting in a reduced consumption of assay reagents and DNA extracts. Microfluidic HT-qPCR platforms offer feasible opportunities to gain more information on quantitative aspects of mosquito biodiversity, which eDNA metabarcoding approaches currently cannot provide, including accurate and fine-tuned spatial and temporal resolution of species distribution assessments and population dynamics. Generally, the fast and highly sensitive detection of specific eDNA targets makes the application of microfluidic technology suitable for real-time monitoring of vector species and implementing timely outbreak management and protection measures. Future assessments should evaluate the use of Digital IFCs for eDNA-based mosquito detection, since this chip format offers the possibility for absolute quantification of eDNA molecules without the need to construct standard curves, making it highly applicable for quantitative eDNA analysis. Additionally, Digital IFCs can improve the eDNA detectability in complex environmental samples with varying concentrations of eDNA molecules and qPCR inhibitors by reducing the qPCR amplification bias and thus increasing the reproducibility, accuracy and sensitivity for reliable and robust species detection.

Conclusions

We demonstrate the potential use of microfluidic HT-qPCR platforms such as the Biomark HD for robust and reliable eDNA-based detection of mosquito species from water samples. Despite the mentioned limitations, these observations are an important contribution to enhance mosquito monitoring efforts alongside conventional and other eDNA methodologies. Contrary to eDNA metabarcoding approaches, the use case of eDNA chips lies in specific tasks such as targeted measures at specific locations (e.g., vector invasion front) or events (e.g., disease outbreaks, eradication attempts), when quantification results can contribute to monitoring measures. We are confident that eDNA chips targeting invasive as well as native mosquitoes have a great potential to be applied more widely in future large-scale monitoring attempts. Yet, continued research and optimization efforts in eDNA sampling and laboratory procedures are needed to realize the full potential of eDNA chips for a reliable and fast detection of mosquitoes for public health protection. One of our main goals is to develop a robust (semi)-quantitative microfluidic-based eDNA mosquito chip targeting all haematophagous culicid species occurring in Western Europe. The surveillance of susceptible species with confirmed or suspected vector competence will be crucial in the near future to allow quick and appropriate measures to protect human and animal health. This new approach will also be useful for large-scale eDNA-based screening to assess mosquito diversity and the invasion progress of invasive mosquito species as well as in pathogen surveillance, as viral and bacterial disease agents can easily be included in future eDNA chips.

Supplemental Information

Supplemental Information 1 Measured Ct-values obtained via qPCR and HT-qPCR used for statistical analysis

Supplemental Information 2 BLAST results of primers and probes used in this study

Supplemental Information 3 QQ plot of measured Ct values obtained via qPCR and HT-qPCR for all tested modified mosquito qPCR assays

Supplemental Information 4 qPCR standard curves for Ae. albopictus, Ae. japonicus and Ae. koreicus with LOD and LOQ

Supplemental Information 5 qPCR inhibition results per filter type for Ae. japonicus and Ae. koreicus at site cemetery WI Igstadt

Ct value shifts of ≥ 1 indicate PCR inhibition.

Supplemental Information 6 Observed Ct values as function of filtrated water volume until clogging of Sterivex® filters of Ae. albopictus eDNA samples measured via qPCR (●) and HT-qPCR (■)

Supplemental Information 7 Sample site coordinates, used filter medium and final filtered water volumes (in mL) per individual eDNA sample

Supplemental Information 8 List of reference DNA samples from selected Culicidae members used for assay specificity tests

Tissue-derived DNA was used for in vitro tests. Positive (+) or failed amplification (−) results are provided with obtained Ct values at 10 ng/μ l measured via qPCR analysis.

Supplemental Information 9 Standard curve formula, efficiency, R2 and LOQ/LOD values of all assays measured via qPCR

Supplemental Information 10 Sequences obtained from eDNA M13 sequencing for Ae. albopictus, Ae. japonicus and Ae. koreicus extracted from positive eDNA sampling sites

Letters A or B indicate the biological replicates from which the sequences were obtained.

Supplemental Information 11 MIQE checklist

We are grateful to Dr. Judith Kochmann, Dr. Fanny Eberhard, Dr. Sarah Cunze and Dr. Antje Steinbrink for helpful discussions and Biogents (Dr. Andreas Rose, M.Sc. Sergej Sperling) for providing Ae. albopictus larvae. We also thank the Friedhofsverwaltung Wiesbaden and Friedhofsverwaltung Ginsheim for the permission to sample on cemeteries. We also thank the lab team of the Centre for Wildlife Genetics (Senckenberg Gesellschaft für Naturforschung, Gelnhausen)—Berardino Cocchiararo, Yvonne Puder and Janis Eurich—for their assistance in eDNA extraction. We would like to acknowledge Caroline Schubert for the assistance in sampling and laboratory work. We also thank the two unknown reviewers for their comments, which improved the manuscript.

Additional Information and Declarations

Competing Interests

Author Contributions

Data Availability

The authors declare there are no competing interests.

Claudia Wittwer conceived and designed the experiments, performed the experiments, analyzed the data, prepared figures and/or tables, authored or reviewed drafts of the article, and approved the final draft.

Chinhda Sharif conceived and designed the experiments, performed the experiments, analyzed the data, authored or reviewed drafts of the article, and approved the final draft.

Isabelle Schöck performed the experiments, authored or reviewed drafts of the article, and approved the final draft.

Sven Klimpel conceived and designed the experiments, authored or reviewed drafts of the article, and approved the final draft.

The following information was supplied regarding data availability:

The Ct measurements are available in the Supplementary File.

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
