# Peer review of "Mosquitoes on a chip—environmental DNA-based detection of invasive mosquito species using high-throughput real-time PCR"

_PeerJ, doi:10.7717/peerj.17782_

## Round 0.1 · original submission · Major Revisions

Dear Dr. Wittwer,

After this first review round, both reviewers indicated the need for major reviews. Therefore, your manuscript may be accepted for publication in PeerJ after it passes through a thorough revision of its contents. Some of the improvements the reviewers requested refer to the issues listed below. Still, other issues may also need to be solved.

The reviewers indicated that improvements are required in the following issues (but minor improvements are also necessary):
a. Issues concerning the aims and background of the study
b. Introduction restructuring, especially metabarcoding
c. Decrease the amount of subsections
d. Revision of tables and figures is necessary
e. Issues related to data validation on more complex water bodies
f. Issues related to the water sampling
g. Issues concerning the structure of the discussion
h. Concerns regarding the amount of replicates
i. Lack of information related to statistical power

As soon as you prepare a new manuscript version, both reviewers will certainly be available to reassess it.

Sincerely,
Daniel Silva
PeerJ Editor

Reviewer 1 ·

Basic reporting

The language in general throughout the text is clear and unambiguous. There are only few formal inconsistencies. I will list these and minor comments concerning word choices, etc. (see below).
The main aim of the proposed manuscript is valuable, other studies could benefit from the effort to transfer and test already validated assays to newer technology. And indeed, the objective to have one assay and analysis for all existing mosquito species would enhance effective monitoring. The long-term goal is nice (could be extended to e.g. to all species in Europe…). Also methodological testing is always useful and as such the filter tests. However, I have some doubts about the presentation of the project’s aims and background and furthermore on the experimental design and scope (for the second, see my concerns below point 2). My concern is that introduction fails to introduce and to explain central concepts and methods (e.g. “eDNA chips”!), to put the study in context and to highlight its relevance. I finally found in the discussion from line 314-330 what should be in the introduction. I recommend to restructure the introduction and to include in the discussion the topics that are missing, in particular a discussion about DNA metabarcoding (and why HT-qPCR should be used instead?) and likely linked to that about quantitative aspects (while mentioned a couple of times, this is not discussed).

I must admit that the article reads slightly technically. The structure of the methods section could be revised to have less subsections, e.g. “Standard curve” and “qPCR inhibition” can be included in the other sections (if these information are necessary). An option could also be to divide the sections: morphological specimen/ eDNA. It might be easier to read with one for “all analyses’ steps for reference specimen” and “eDNA samples with: extraction, qPCR and HT-qPCR”.
And I’m afraid that in my opinion the figures and tables are not well chosen here and the way the results are represented should be revised, in particular since the used markers have been validated before in other studies. The main objectives of this study being a) comparison between qPCR and HTqPCR and also of the b) the filters, only Fig. 3, 4 and 5 and/or Table 4 are relevant to show.
Fig. 1: The map is not appropriate to indicate the sampling spots. It is way too large and the sampling dots have the size of cities... But the other question is if a sampling map in the main text is interesting given that the detection of invasive mosquitoes in the field is no real interest in this study. Table S1 could be enough.
Fig. 2: The qPCR have been tested in the Schneider et al 2016 paper already with assessment of LOD and LOG. Not sure this figure is relevant for the main article, could be Supplementary material.
Table 1 is not necessary, all used primers are referenced in the text and primers details given in the corresponding studies.
Table 2 since values are reported in the text, this table is unnecessary too or could be Supplementary material. Furthermore, these markers have been evaluated previously and their specificity has been validated.
Table 3: see comment Table 2.

Abstract
Line 32: complementary in stead of complement?
Line 36: I suggest to add abbreviations HT-qPCR and qPCR in brackets
Line 42: Ct values measured by qPCR in stead of measured Ct values via qPCR
Line 42: Moreover, … Missing comma but also ambiguous sentence, please rephrase and rethink. (I’m aware why this is phrased like this but given the pre-PCR step, it is also debatable if Ct values should be compared this way.)
Line 53: delete attempts
Line 54: eDNA Mosquito Chip => either all italic or small letters for “chip”, once more line 54 and please check throughout the manuscript
Line 56: delete susceptible

Introduction
I suggest to review the text from line 66-94 for content ambiguities. E.g. if active monitoring is sampling of eggs, why fully adults needed? Active/passive monitoring not well defined here and/or arguments not well structured.
Line 100: The Ficetola et al 2008 paper employed eDNA techniques to detect frogs not fish.
Line 109: “DNA metabarcoding or other traditional survey methods”? DNA metabarcoding is no traditional method.
Line 114: Introduce better and explain the functioning of “eDNA chips”. This is a basic concept for this article and it is explained nowhere how it works.
Line 122: “different filter media” also needs more explanation, e.g. filters with various pore sizes. How does the procedure works? At this moment of the manuscript the reader does not know.

Materials & Methods
Line 127: delete Site selection,
Line 135: Information missing for Biogents. Throughout the text, please make sure that you are formally consistent, see e.g. lines 156 and 162 concerning Qiagen.
Lines 138ff: It is difficult to follow and understand the different filtering procedures, structure…
Line 147: “sealed…to an icebox”?
Line 162: Blood & Tissue, not + see line 156.
Line 166: DNA barcoding
Line 180: NCBI not the Pubmed (to be consistent throughout the text)
Line 183: already existing => developed and tested in previous studies
Line 186: delete “use on the”?
Line 191: Please revise the presentation of the different steps, e.g. consider deleting “of denaturation; annealing/ elongation”. Also were biological or technical replicates used?
Line 204: throughput not throughout
Line 210: Is there a step missing (like 7 minutes on 72°C) or is this one not there on purpose?
Line 213ff: The use and numbers of replicates is still not clear to me…
Line 228: “with the given qPCR protocol”. Does this mean 1 sample per site tagged and then qPCR and HTqPCR? Or 1 sample with qPCR for all 3 assays?
Line 248: It is not necessary to state explicitly.

Results, discussion, conclusions
Line 257: rephrase please sentence “Assuming…”
Line 317: if the term “eDNA marker” is used, it should be defined or otherwise rephrased, e.g. makers used for eDNA-based assessments…

Experimental design

I’m not sure I got every detail about the study design. As far as I understood there were no true positive controls in the field; regional previous assessments are debatable given the breeding biology of the three species in question (small, and also often temporal, water containers). Not sure why these were included then, supposedly to have “real” eDNA samples but this should be clearly stated somewhere. A schema/visualisation of which sample has been analysed how (filters, etc.) might be helpful.
Regarding the small sample size, I am aware that this is a preliminary study. But it raises nevertheless several questions. On the one hand a lot of steps reported here are not new since the primers have been validated and tested before for qPCR and this does not absolutely need to be published again in this form. On the other hand the sample size is very limited and except the Ae. albopictus samples, samples do not represent real positive controls (i.e. regional assessment vs. local breeding containers) and no real monitoring of mosquitoes is involved. We learn that filter pore sizes do not affect the results and that the primers can successfully be used on a newer HT-qPCR system. I am a lot for the idea to transfer existing techniques to newer systems and to test this, however, I think it would be beneficial for the scope of this study to include higher sample numbers (from the lab or from the field), i.e. to extent the current dataset. Or to include the current preliminary tests into a publication that focuses on the monitoring part which is probably planned with the now tested HT-qPCR system.
In general, the recommendation in eDNA studies is to use three technical replicates but since I have not understood which are biological and which technical replicates, I’m not sure about this point here.

It is problematic to refer to unpublished work (Wittwer & Nowak 2023) without giving enough details to replicate the analyses. With the presented information, the study cannot be replicated. My recommendation is to include more details about the methods as long as the other study is not published or wait until it is...

Validity of the findings

While the aim of the study is well-founded, it has some limitations regarding the design and also partly the novelty of the findings. The manuscript is solid but should be revised in order to emphasise the contextual background and arguments, as well as to expand the scope of the discussion.

Reviewer 2 ·

Basic reporting

a. One major concern with this study is the lack of a validation in water bodies of more complex ecological structure, i.e. is there a risk of false positives in permanent water bodies such as ditches/lakes/wetlands with more complex water chemistry and ecological communities (i.e. high abundances of chironomids, crustaceans etc). Did the authors at least test in silico for common German non-culicid macroinvertebrates that are highly related such as Chironomidae and Dixidae? If not, how do the authors propose to prevent false positives, and misjudged interventions?

b. I couldn’t find a description of how the water samples were collected in the section on eDNA sampling. E.g. was vegetation and algal film evaded to limit collection of algae and humic acid. Which part of the water column was sampled, and was this informed by mosquito ecology? How were the filtration volumes determined and adjusted? Namely, a volume of 200ml is reported, but later in the paper filtration volumes are evaluated. Did the authors test and verify the decontamination by filtering deionized water after cleaning?

c. I find the current structure of the discussion somewhat confusing. Currently, the authors give the general patterns, elaborate on broader applicability, then summarize their findings and finally give the limitations. Thereafter the conclusion once again gives the general patterns. I propose that the authors start with their general patterns, summarize their findings, mention the limitations and move the broader applicability (e.g. r317-328) to the conclusion or end of the discussion.

Experimental design

Regarding the comparison of filter media, I have difficulty determining what the authors are trying to compare, the effect of pore size or the effect of the extraction protocol, as both are done simultaneously. Why were these three media and corresponding extraction protocols chosen, and why was only one filter type used for Ae. albopictus? To my understanding Sterivex filters are (often) made of PES, so what difference did the authors expect between the Sterivex and PES filters?

Validity of the findings

a. I fail to understand why the authors use so few technical replicates, i.e. 1 for the estimation of the PCR efficiency, and 1-2 for the collected samples, and instead chose to explore so many variables. Using >3 technical replicates is generally accepted as the norm given the stochastic nature of qPCR (Keenum 2022, 10.1080/10643389.2021.2024739; Svec 2015, 10.1016/j.bdq.2015.01.005). Similarly, why did the authors use different amounts of replicates for the three species, i.e. two for Ae. albopictus and a single for both Ae. Japonicus and Ae. Koreicus?

b. I find the notion ‘all tested filter types were well-performing’ in the discussion misleading, as the filters performed well in three instances spread over two locations. I would appreciate it if the authors explicitly stated the limitations arising from the limited number of samples. Furthermore, I missed any mentions on the statistical power, making it impossible to assess whether the findings are justifiable, especially combined with the low sample size and lack of technical replicates.

Additional comments

Minor points:
- r36 and throughout, it is not explicitly stated that the filter media also differ in pore size.
- r75 It is unclear whether all 3 species were collected using both methods, or whether some were collected using active sampling and others using passive sampling.
- r80 Given that this seems to be the reason for selecting these species for this study, please state, if possible, with references, that these species are the main concern of your vector control or other reasoning for selecting these three specifically.
- r81 Needs a reference.
- r83 ‘living or dead’ may be omitted.
- r90 I find ‘very small’ a bit awkward given the repetition. Maybe replace with ‘early developmental stages’.
- r104 ‘only for a short period of time for’ may be omitted.
- r105 ‘with mosquito eDNA being detectable for at least 25 days’ is a bold claim given the single reference. Please rephrase to 'with one study reporting mosquito eDNA being detected up to 25 days'.
- r128 Please state the number of sites.
- r158 Please state which buffer.
- r163 Please state which buffer.
- R167 Were the larvae processed similarly to the reference specimen? Were these decapitated also?
- r172 ‘extraction’
- r239 The HT-qPCR data seems to be non-normally distributed, with more extreme values than expected. Why were t-tests used instead of a Mann-Whitney U?
- r257 This sentence reads somewhat awkwardly.
- r311 Please elaborate on how the modified protocol compared to the original sensitivity.
- r337 ‘the amplification of eDNA samples via HT-qPCR can often be pushed over LOD’ needs a reference.
- r343 ‘On the other hand’ may be replaced by ‘Similarly’.
- r344 ‘Main reason’ may be replaced by ‘The main reason’
- r351 ‘little is known of eDNA release patterns of mosquitoes’, there is a relevant article the authors may want to consider citing: Sakata 2022 (10.1371/journal.pone.0272653).
- r365 Wouldn’t varying abundances overshadow the effect of varying release patterns?
- Table 4. Grouping Ae. albopictus with the other species insinuates that this species was assessed using the same methods (all filter types, same amount of replicates). For clarity I propose to put sample 5-14 in a separate table.

---

## Round 0.2 · accepted · Accept

Dear Dr. Wittwer,

I am pleased to accept your manuscript for publication in PeerJ! Congratulations!

Sincerely,
Daniel Silva

Reviewer 1 ·

Basic reporting

Well done, the authors have greatly improved the manuscript. The article reads very smoothly now and all the issues have been addressed. In particular, the introduction and discussion contain the necessary background information and references and Figure 1 is a very nice complement to the text.

Some very minor propositions:
Line 76: twice “either”
Line 79: Alternatively, (comma)
Line 198: “scale up for large-scale surveillance” => word repetition intended?
Line 238: not sure this needs to be repeated here: from Hill et al. (2008), Van der Vossenberg et al. (2015) and Schneider et al. (2016)
Line 240: eDNA Chip => eDNA chip, according to the authors reply
Line 453: permanent => permanently
Line 892: concentraction => concentration
Line 910: High numbers of shedded juvenile exuviae can serve as important source for eDNA => maybe rephrase
Line 928: Most currently available eDNA markers primarily target invasive species => somehow repetitive: most and primarily
Line 941: for a robust species detection => for robust species detection?
Line 953: of the huge metabarcoding data sets => of huge metabarcoding data sets

“All in all” is maybe used too often throughout the text.

Experimental design

Great that technical replicates have been added.

Validity of the findings

The study’s relevance and scope have been justified in this version.